biomaterials/materials science

electrospinning, mosquito, repellent, emulsion, micro-capsules

**Author for correspondence:**
Lucy Ciera
e-mail: lucyciera2@gmail.com

# Electrospinning repellents in polyvinyl alcohol-nanofibres for obtaining mosquito-repelling fabrics

Lucy Ciera[5], Lynda Beladjal[2], Lieve Van Landuyt[1], David Menger[3], Maarten Holdinga[3], Johan Mertens[2], Lieva Van Langenhove[1], Karen De Clerk[1] and Tom Gheysens[1,4]

[1]Textile Department, Ghent University, Technologiepark, Zwijnaarde, Belgium
[2]Biology Department, Ghent University, Terrestrial Ecology Unit, Gent, Belgium
[3]Laboratory of Entomology, Wageningen University, Wageningen, The Netherlands
[4]Department of Organic and Macromolecular Chemistry, Ghent University, Ghent, Belgium
[5]The Technical University of Kenya, Nairobi, Kenya

LC, 0000-0002-6474-2770; LB, 0000-0002-5755-7235

Recently, the use of repellents for preventing the transmission of mosquito-borne diseases is getting increasingly more attention. However, most of the current repellents are volatile in nature and must be frequently re-applied as their efficacy is only limited to a short period of time. Therefore, a slow release and abrasion-resistant mechanism is needed for prolonging the protection time of the repellents. The focus of this study is on the direct micro-encapsulation of repellents from an emulsion and integration of already encapsulated repellents into nanofibres via electrospinning. Different repellents were electrospun in polyvinyl alcohol (PVA) nanofibrous structures, namely *p*-menthane-3,8-diol micro-capsules, permethrin, chilli and catnip oil. The repellents were successfully incorporated in the nanofibres and the tensile properties of the resulting samples did not have a significant change. This means that the newly created textiles were identical to current PVA nanofibrous textiles with the added benefit of being mosquito repellent. Principally, all incorporated repellents in the nanofibrous structures showed a significantly reduced number of mosquito landings compared to the control. Consequently, the currently described method resulted in a new and very effective repelling textile material that can be used in the prevention against mosquito-associated diseases.

# 1. Introduction

Mosquitoes are vectors for many diseases like yellow fever, dengue, Nile fever, encephalitis and malaria, transmitted through mosquito bites. In 2017, there were an estimated 435 000 deaths from malaria globally [1]. The lack of an effective vaccine against mosquito-borne diseases makes the prevention of mosquito bites the most important strategy in countering the transmission of these diseases [2,3]. The most effective and most commonly used insect repellents for preventing mosquito bites present on the market today are synthetic in nature and include DEET (*N*, *N*-diethyl-3-methylbenzamide) and picaridin [2,4,5]. However, some of these synthetic repellents are associated with some harmful and toxic effects, some resulting in allergic reactions and damage to the nervous system [6,7]. Moreover, some mosquitoes have become resistant to the current synthetic repellents and environmental concerns have prevented their effective use [3,5].

Therefore, there is a large market for alternative repellents. Thus, lemon eucalyptus, chilli and catnip oil were used in this study besides permethrin. *p*-menthane-3, 8-diol (PMD) is the active compound of lemon eucalyptus oil and is extracted from the twigs and leaves of *Eucalyptus citriodora* of the Myrtaceae family [5,7]. The repellent properties of PMD consist of masking the environmental cues used by mosquitoes to find their host [4,8,9]. Chilli essential oil, another biological repellent, has capsaicin and dihydrocapsaicin as its active compounds. Chilli oil gives a hot flavour which has a significant anti-feedant and lethal effects on mosquitoes, dogs, rabbit, cats, birds, cotton pests, etc. [6,10]. Likewise, catnip essential oil, with nepetalactoneas as the main active compound, has a pungent, minty distinctive smell that repels mosquitoes, spittle bugs, termites, ticks, spiders and cockroaches [9,11].

Repellents are mostly applied directly onto the skin or impregnated on fabric surfaces. However, due to their volatile nature, their efficacy is only limited to a short time period and the repellents need frequent re-application. Also, their durability and resistance to abrasion is questionable when coated on the fabric surface. Additionally, some repellents like permethrin and chilli oil cause skin and eye irritation when in direct contact and must be used with caution [12]. Direct micro-encapsulation of these repellents in nanofibres during emulsion electrospinning or integration of already micro-encapsulated repellents in the nanofibres would offer a possible solution to these problems. Firstly, the micro-capsules would provide a controlled delivery mechanism with longer protection times. Secondly, using electrospun nanofibres will lead to high functional textiles with maximum repellency due to the huge surface areas resulting in optimal evaporation of the repellent when the containing capsules are broken. Lastly, micro-encapsulation prevents repellents from being in direct contact with the skin, therefore reducing skin irritation and/or allergic reaction. An additional advantage of using electrospinning is that it is a fast and cheap technique for producing fibres at the nanometre scale ($\Phi$: 10–800 nm) [13]. Therefore, in this paper, we compared both techniques, emulsion and micro-capsule electrospinning, in their ability to integrate repellents into nanofibrous structures for textile applications.

Nanofibres have been successfully electrospun from polymers like polyamides (PA), polyester (PE) and polyvinyl alcohol (PVA) [13,14]. However, most of these polymers, like PA and PE, can only be dissolved in toxic solvents and are non-biodegradable, making them non-environmental-friendly and harmful to use. Therefore, PVA was chosen for this study as it is water soluble hence there is no need for toxic solvents for dissolution, it is easy to process and is biodegradable [14]. Moreover, PVA is used in a wide range of applications in the medical, cosmetics and textiles sectors.

In this paper, electrospinning was optimized to incorporate different concentrations of permethrin, PMD, chilli and catnip oil repellents in PVA nanofabrics via micro-encapsulation for PMD or direct PVA-repellent emulsion spinning for permethrin, catnip oil and chilli oil.

Because the selection of materials for textile and industrial applications strongly depends on their strength which is determined by the mechanical properties, these properties were tested on the newly created fibrous materials. The mechanical properties like tensile strength, Young's modulus and elongation at break, which determine the strength of the material, were analysed with a Favimat tensile testing machine. On the other hand, the morphology of the resulting nanofibres was determined with scanning electron microscope (SEM) which was aimed at testing the electrospinning set-up and parameters. Moreover, to determine the presence of incorporated repellents in the PVA nanofibrous structure, Raman spectroscopy was used. Finally, a vertical landing bioassay was used to investigate mosquito landings on the resulting nanofibres which determined the effectiveness of this novel technique in developing mosquito repellent textiles.

Here, we show for the first time a novel functionalized textile material that significantly repels mosquitoes of the species *Anopheles gambiae* (and possibly others), responsible for spreading malaria among humans. These PVA-nanofibre fabrics can be easily integrated into existing textiles to develop

a new generation of mosquito-repelling textiles. Moreover, these fabrics possibly have slow-release capabilities due to the presence of micro/nano-capsules containing the repellents, unseen in other textile materials.

Many compounds, both synthetic chemicals and natural like plant-derived essential oils, have been used against flies, dust mite, cockroaches, stick insects, ticks, bedbugs, bees, head lice, wasps, ants, horse flies, gnats, chiggers and other arthropods [1,2,9,12]. Therefore, many more repellents than those tested here can be incorporated in textiles either by direct micro-encapsulation of repellents from an emulsion or integrating already encapsulated repellents into nanofibres. This can provide an entirely new class of repelling textiles that not only protect against mosquitoes but can also protect against other insects, therefore increasing the applications of the textile materials. It is expected that the novel repellent textile fabrics presented here will become key in the reduction in mosquito-borne diseases in tropical and subtropical regions.

# 2. Material and methods

PVA (Mowiol 40–88, MW 205 000) was supplied by Sigma Aldrich. Chilli and catnip essential oil, as well as micro-capsules ($\Phi$: $5 \pm 1\,\mu m$) made from melamine formal resin containing $p$-menthane-3, 8-diol (PMD) were supplied by Devan chemicals (Belgium). Permethrin was supplied by Utexbel (Belgium).

## 2.1. Preparation of the spinning solutions

The spinning solution was prepared by dissolving the required amount of PVA in 20 ml distilled water to obtain an 8 wt% solution. This solution was gently stirred with a magnetic stirrer for 6 h at room temperature ($21 \pm 1°C$). For electrospinning the micro-capsules, this solution was used without further additions or manipulations. However, for emulsion electro-spinning, repellents were pipetted into the prepared PVA solution to obtain 0 (control), 2, 4, 6, 8, 10, 12, 14 and 16 wt% concentration of repellents to PVA solution. The solutions were then sonicated for 30 min using a Branson 1510 Sonicator (USA) to obtain a uniform emulsion and then electrospun immediately to avoid phase separation.

## 2.2. Electrospinning of PVA and PVA/mosquito repellent nanofibrous structure

The prepared solutions were electrospun into nanofibres using a mono-nozzle electrospinning set-up as described in [15]. In short, the polymer solution was pumped from a 20 ml plastic disposable syringe into a 15 cm long stainless steel needle with an inner diameter of 1.024 mm (Sigma-Aldrich) connected to the positive terminal. The flow rate of the spinning solution was controlled with a syringe pump (KD Scientific, USA), while the distance between the needle tip and the collector was adjusted with a laboratory jack. The high voltage was sourced from a Glassman High Voltage Series EH30P3 supply unit and the nanofibrous structures were produced with a power range of 12–23 DC kV. A plate connected to the negative electrode of the power supply was covered with an aluminium foil and was used as the collector. The flow rate of the spinning solution was set at $1\,ml\,h^{-1}$ and the distance between the needle tip and the collector was 14 cm. The nanofibres were electrospun at room temperature of $21 \pm 1°C$ and a relative humidity of $40 \pm 5\%$.

## 2.3. Characterization of electrospun samples

Because only emulsion up to 8 wt% catnip oil could be successfully electrospun, samples with 8 wt% of permethrin, chilli and catnip oil emulsion were used in the characterization for comparison. In the case of PMD micro-capsules, the characterization was done for a range of samples up to 16 wt%.

### 2.3.1. Scanning electron microscopy

The morphology of the resulting nanofibres was studied using an SEM (Joel Quanta 200 F FE-SEM). The samples were placed on an SEM stub and given a 5 nm gold coat using a sputter coater (Balzers Union SKD 030). The SEM micrographs were made at 20.0 kV with a spot size of 4–6 nm, a dwell of 3000 µs and a working distance of 10 mm. To determine the average fibre diameter and thus the consistency/reproducibility of our electrospinning, 50 diameter measurements per sample (20 000× magnification) were taken using Cell D software (Olympus).

### 2.3.2. Raman spectroscopy

Raman spectroscopy was used to spectrally confirm the presence of the repellents in the resulting nanofibrous structure. The tests were run in a Perkin-Elmer Spectrum GX 2000 spectrometer with a Raman beam splitter, a diode-pumped YAG infrared laser (1064 nm and laser power of 800 Mw) and InGaAs detector. The measurements were recorded in the back-scattering mode (180° excitation optics) with 128 scans, a resolution of 4 cm$^{-1}$, a data-interval of 1 cm$^{-1}$ from 3500 to 100 cm$^{-1}$. Prior to obtaining the spectra, samples were peeled off from the aluminium foil and aligned on a sample holder.

### 2.3.3. Favimat tensile testing

To evaluate the mechanical properties of the resulting nanofibrous structures, a longer electrospinning collection time of about 1 h for each sample was used in order to produce thicker nanofibre mats suitable for the analysis. The mechanical analyses were performed with a FAVIMAT-ROBOT (Textechno, Germany). The measurements were carried out with $3 \times 30$ mm sample strips carefully cut out with scissors of the corresponding electrospun material. Prior to testing, all samples were weighed with an average weight of $771 \pm 1$ mg to ensure an equal thickness between samples. The gauge length and test speed were set at 20 mm and 20 mm min$^{-1}$, respectively. An average of 20 strips was tested per sample.

## 2.4. Characterizing the repellency of the integrated mosquito repellents

### 2.4.1. Mosquitoes

*Anopheles gambiae s.s.* mosquitoes were used in this study. *Anopheles gambiae s.s.*, a member of the homonymous species complex comprising six recognized sibling species, is the most anthropophilic malaria vector worldwide [16,17]. Within the *A. gambiae* (Diptera: Culcidae) complex, *An. gambiae sensu stricto (s.s)* is the most specifically adapted to humans and has the highest malaria vectorial capacity [18,19].

The *A. gambiae s.s.* mosquitoes were reared in climate chambers at the laboratory of entomology of Wageningen University (The Netherlands), while the original population was collected in Suakoko (Liberia). The mosquitoes were kept under phase shifted: scotophase of 12 : 12 h at $27 \pm 1$°C and relative humidity of $80 \pm 5\%$. Adult mosquitoes were kept in $30 \times 30 \times 30$ cm gauze wire cages and had access to human blood on a Parafilm® membrane every other day and a 6% glucose solution in water (ad libitum). Eggs were laid on a wet filter paper and then placed on a plastic tray with tap water for emergence. Larvae were fed on Liquifry® No. 1 (Interpet, UK) for the first 3 days and then with TetraMin® baby fish food (Tetra, Germany) until they reached the adult stadium. Pupae were collected from the trays using a vacuum system and placed into a plastic cup filled with tap water for emergence.

The mosquitoes intended for these experiments were placed in separate cages as pupae, where they had access to a 6% glucose solution but received no blood meals. The day before the experiment, 5- to 8-day-old female mosquitoes were placed in release cages with access to tap water in cotton wool until the experiment. Both laboratory experiments took place during the last 4 h of the scotophase, a period during which *A. gambiae s.s.* females are highly responsive to host odours. The environmental conditions of the room where the repellence tests were run were continuously monitored using a Tinyview® data logger with display. The temperature was maintained at $26 \pm 1$°C.

### 2.4.2. Bioassay

A vertical landing bioassay was used to determine the possible repellence of the new incorporated repellence fabrics. The assay consisted of a cubic cage with steel gauze on three sides and Perspex on the other three sides. Underneath the gauze at the bottom of the cubic cage, a warmed circular plateau (Ø 15 cm) was positioned on top of which moist filter paper was applied and an attractive odour blend was released. The attractive odour strips were prepared by cutting nylon strips measuring 26.5 cm, from panty hoses (90% PA, 10% spandex, Marie Claire®) (figure 1).

The strips were then impregnated with attractive compounds by bringing them into an Eppendorf tube containing 1 ml of solution as described in [21]. Thereafter, the strips were stored overnight in a refrigerator at 4°C and later hung up for half an hour under a fume hood to allow excess fluids to leak out. Finally, the strips were immediately used or packed in aluminium foil and stored at 4°C in a refrigerator until they were used.

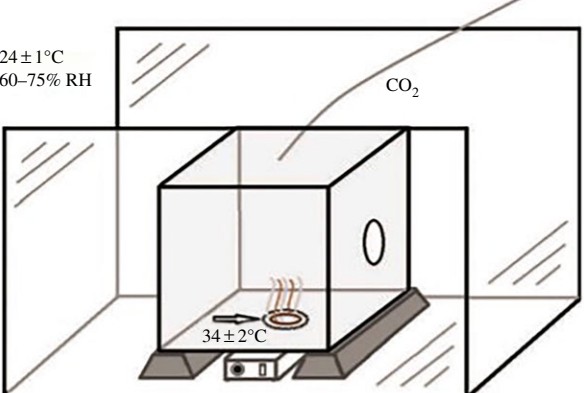

**Figure 1.** Schematic of the repellent bioassay, showing the flight chamber containing the assay cage and the position of the circular landing platform (arrow) emitting five-component attractant blend and moisture, on which the repellent-impregnated PVA nanofibres were placed [20].

The temperature at the gauze was kept at $35 \pm 2°C$, comparable to the temperature of human skin. This resulted in a high attraction of female mosquitoes to the warm area on the bottom of the cubic cage where they would land and probe with their proboscis through the gauze in search of a blood-host.

The resulting PVA-nanofibre mats ($9 \times 9$ cm sized) incorporated with different concentration of PMD, permethrin, chilli and catnip oil were stored in sealed plastic packaging under the same conditions as the nylon strips before usage. In the first day of the experiments, a circular hole of 5 cm Ø was cut into the middle of the sample to allow warm, moist air with attractive volatiles to pass through the sample. The sample was then placed at the bottom of the cage over the warm attractive area. A control (blank PVA nanofibres) and PVA nanofibres incorporated with different concentrations of the repellents were tested. Each sample was replicated 10 times; tests were randomized within each repetition and spread over 5 different days.

The repellency of the samples was measured by releasing a group of 10, 7-day-old female mosquitoes into the cage. After 2 min of acclimatization time, landings made on the nanofibres or within the circular area of the sample were counted for a period of 8 min. A landing was defined as the total period for which a mosquito maintained contact with the landing platform. Walking/hopping around on the landing plateau as well as short take offs (less than 1 s) immediately followed by landing again were included in one landing. A new landing was recorded when a mosquito had left the plateau for more than 1 s before landing again. Landings shorter than 1 s during which no probing took place were ignored.

## 2.5. Statistical tests

The two-sample Kolmogorov–Smirnov statistical test was used to analyse the effect of adding the repellents to the nanofibrous structure. An effect is significant when the $p$-value is smaller than 0.01. Comparisons were made between all treatments in addition to the comparisons between treatments and the control.

# 3. Results

## 3.1. Morphology of the fibres

The fibre morphology of the electrospun samples of the reference blank PVA nanofibrous structure (figure 2a) looked similar to the sample with added PMD micro-capsules (figure 2b). The micro-capsules were covered by the nanofibres integrating them into the fabric, not changing the nanofibre morphology.

In the case of permethrin, chilli and catnip oil emulsion electrospinning, a smooth and uniform fibre surface was observed for the reference blank PVA nanofibres (figure 2c). However, PVA nanofibres loaded with permethrin (figure 2d) and catnip oil (figure 2e) showed bead-like structures along the fibre axis (figure 2 (arrows)), while fibres loaded with chilli oil showed fibres with pores (figure 2f (arrows)).

Incorporating PMD micro-capsules had a significant effect on the average diameter of the nanofibres ($p$-value: 0.0001) (figure 3a). A decreasing trend was seen in the diameter of the nanofibres upon adding

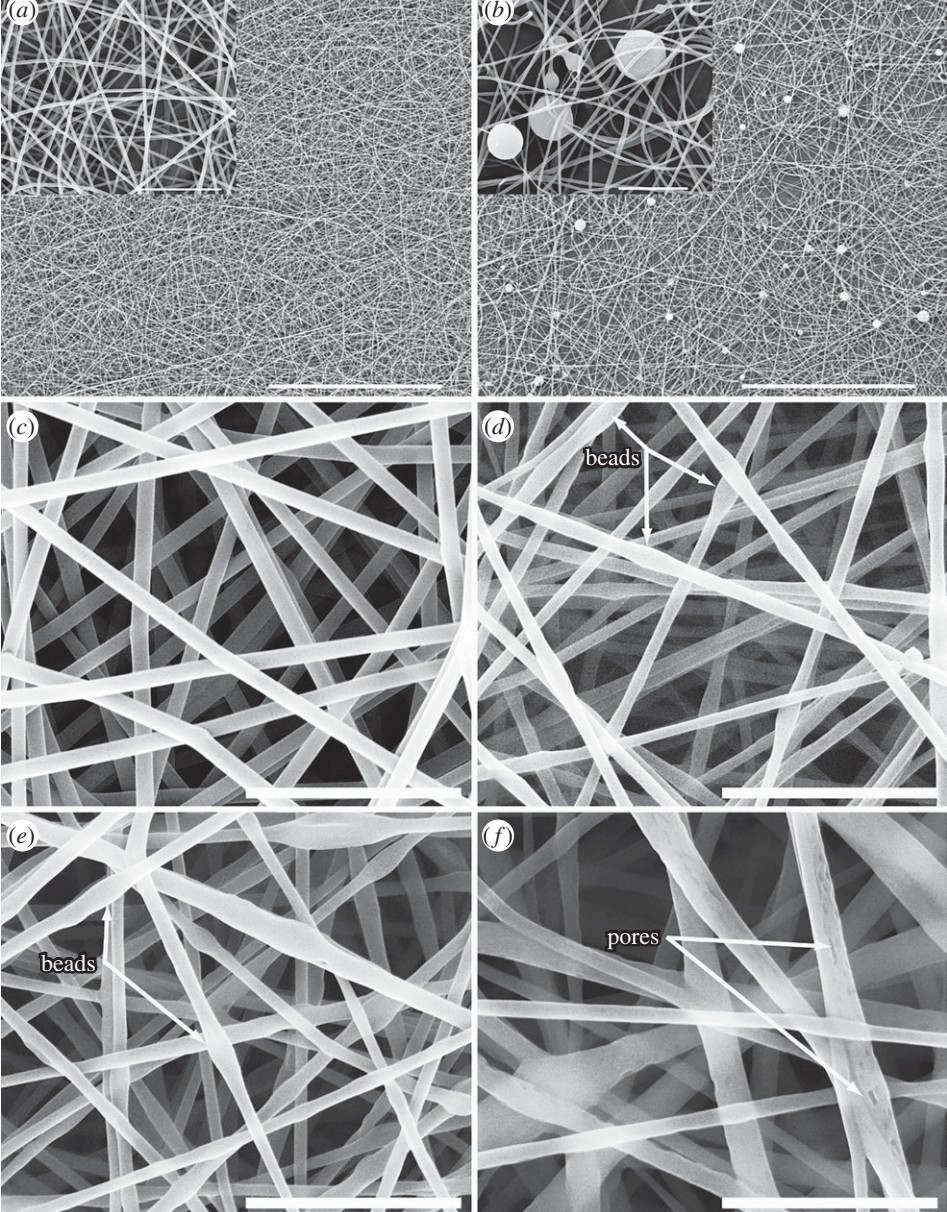

**Figure 2.** SEM micrographs of PVA nanofibres: (*a*) blank PVA nanofibres, (*b*) PVA nanofibres incorporated with PMD micro-capsules, (*c*) blank PVA nanofibres, (*d*) emulsions of incorporated permethrin (8%), (*e*) catnip oil (8%) and (*f*) chilli oil (8%). Scale bar inset pictures: 100 μm for (*a*,*b*); 10 μm for (*c*–*e*); 5 μm for *f*. [21].

more micro-capsules, with 16% PMD micro-capsules showing the smallest average diameter and a large size distribution (386 ± 55 nm) against the blank (408 ± 67 nm).

The results presented in figure 3*b*–*d* show that the emulsion spun nanofibres with permethrin, catnip and chilli oil had a highly significant effect on the average diameter of the resulting nanofibres (all $p < 0.0001$). An increasing concentration of permethrin resulted in a decreasing fibre diameter (figure 3*b*), while the diameter of fibres loaded with chilli and catnip oil increased with increasing concentration of the repellents (figure 3*c*,*d*). Additionally, there was a big variability in the fibre diameters for fibres loaded with chilli in contrast to the others with increasing concentrations.

## 3.2. Mechanical properties

As shown in figure 4, introducing PMD micro-capsules, permethrin, chilli and catnip oil into PVA nanofibrous structures did not significantly change the tensile strength, Young's modulus and elongation at break of the resulting nanofibrous structure. Although not significant, samples incorporated with catnip did show a slight decrease in mechanical properties.

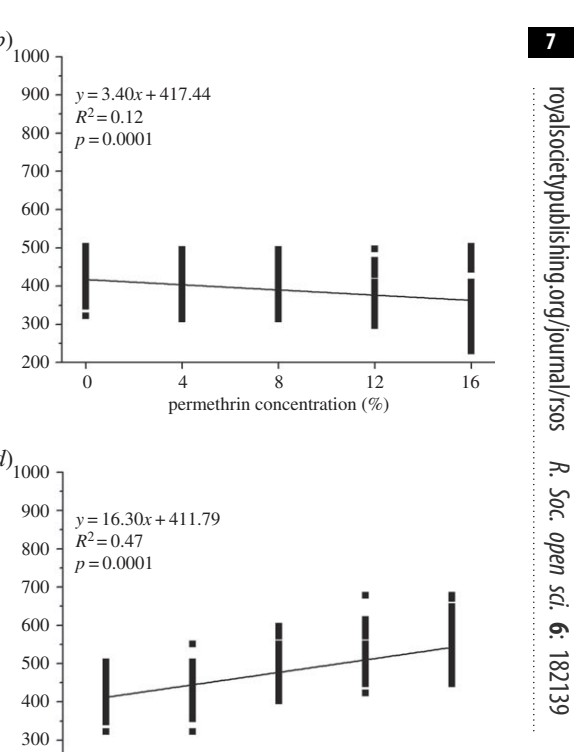

**Figure 3.** Linear regression model showing the variation in nanofibre diameters incorporated with different concentrations of PMD micro-capsules (*a*), permethrin (*b*), chilli oil (*c*) and catnip oil (*d*) [21].

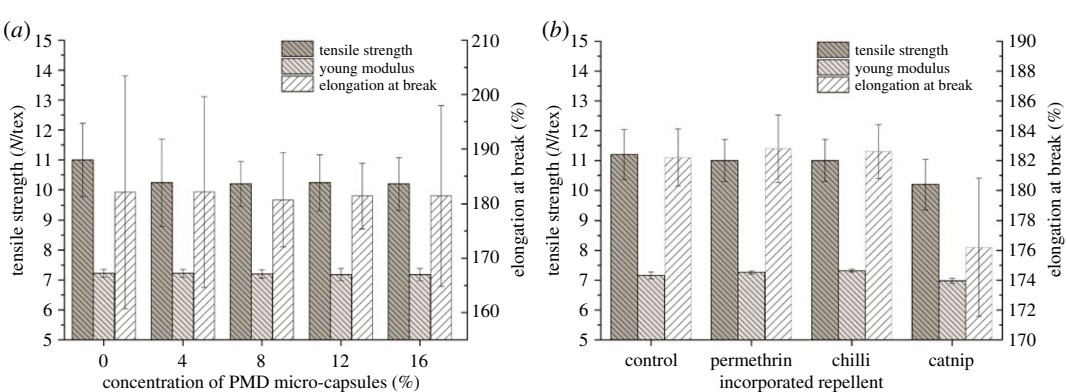

**Figure 4.** Mechanical properties of PVA nanofibres incorporated with different concentrations of PMD micro-capsules (*a*), permethrin, chilli and catnip (*b*) [21].

## 3.3. Raman spectroscopy

Raman spectroscopy was used to determine the presence of incorporated repellents like PMD micro-capsules, permethrin, chilli and catnip oil in the PVA nanofibrous structure. The Raman vibration spectra of pure PVA (black) and incorporated repellents (red) were initially recorded separately whereafter spectra of PVA nanofibrous structures incorporated with each repellent were taken. From these spectra, a subtraction spectra (blue) was distilled by subtracting the pure PVA spectra from the spectra of the PVA structure incorporated with the repellent (figure 5).

The spectrum of blank PVA showed characteristic peaks at 848, 2906, 2911, 1437, 1438 and 1439 cm$^{-1}$ (figure 5 (black)).

The PMD micro-capsules spectra showed sharp peaks at 759 and 1438 cm$^{-1}$ (figure 5*a* (red)). The subtraction spectra (blue) and the spectra for PVA with PMD incorporated micro-capsules (green/yellow) both showed peaks at 759 and 1438 cm$^{-1}$. Moreover, the peak at 759 cm$^{-1}$ increased in intensity with

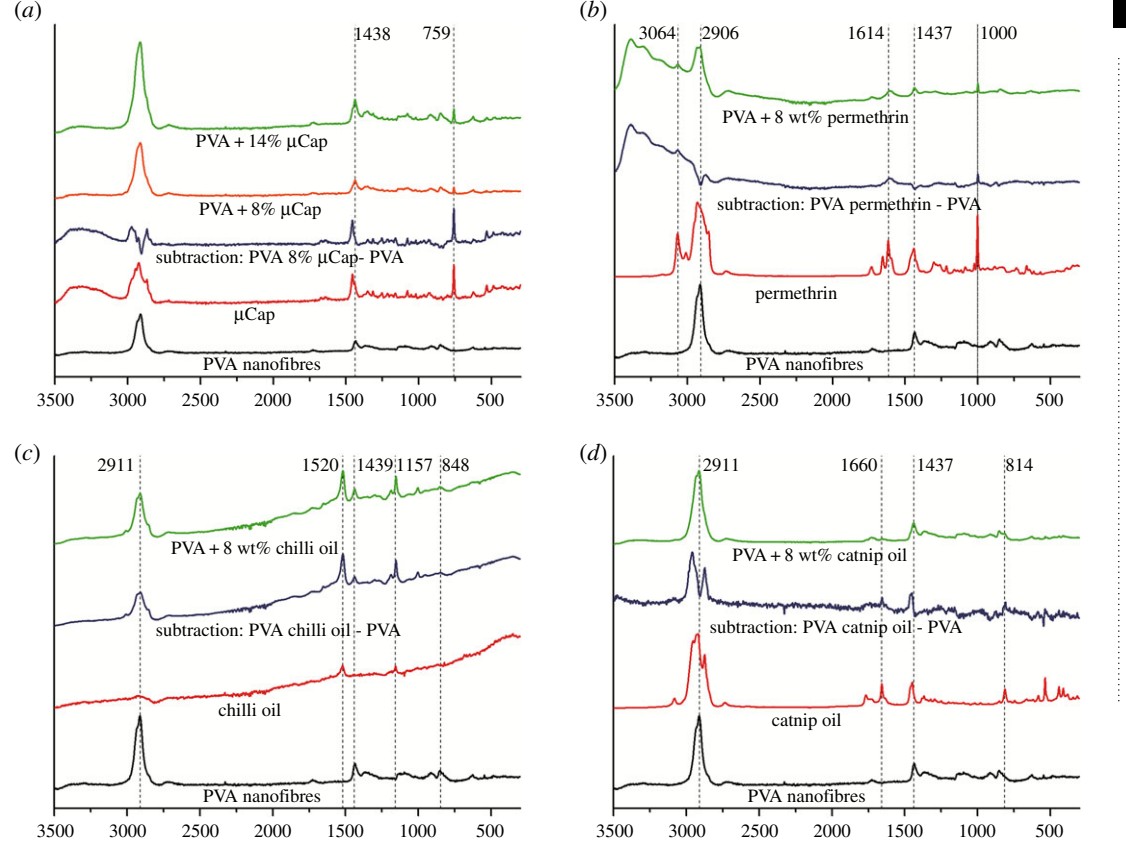

**Figure 5.** Raman spectra of PVA nanofibrous structure incorporated with PMD micro-capsules (*a*), permethrin (*b*), chilli oil (*c*) and catnip oil (*d*) [21].

increasing micro-capsule concentration (figure 5*a* (green and yellow)). This implied that the peak at 759 cm$^{-1}$ is a representative peak for the presence of micro-capsules in the resulting nanofibrous structures, confirming the presence of micro-capsules in the spun PVA nanofibrous structures.

For the pure permethrin spectra (figure 5*b*), four characteristic peaks at 3064, 1614, 1437 and 1000 cm$^{-1}$ could be observed. When subtracting the spectra of pure PVA from PVA with permethrin, the same characteristic peaks, namely 3064, 1614 and 1000 cm$^{-1}$, were present as in pure permethrin (not found in PVA) and therefore can be used for the identification of permethrin in PVA nanofibrous structures. The spectra for the sample of PVA incorporated with permethrin showed the peaks for both PVA and permethrin, namely at 1614 and 1000 cm$^{-1}$ for permethrin, and 1437 for PVA, confirming the presence of permethrin in the spun PVA nanofibrous structure.

Raman spectra for the sample with incorporated chilli oil are presented in figure 5*c*. The spectrum for pure chilli showed two characteristic peaks at 1520 and 1157 cm$^{-1}$. In contrast with permethrin, chilli oil had no common characteristic peaks with pure PVA and so both peaks were used as indicators for the presence of chilli oil in the PVA nanofibrous structure. The sample of PVA with incorporated chilli oil had a spectra with peaks associated with both chilli oil and PVA, indicating that the chilli oil was integrated into the spun PVA nanofibrous structures.

The spectra of PVA with incorporated catnip are presented in figure 5*d*. The spectra for pure catnip oil showed four characteristic peaks at 2911, 1660, 1437 and 814 cm$^{-1}$. Like with permethrin, catnip and PVA share common bands at 1437 and 2911 cm$^{-1}$. The spectra of the PVA with incorporated catnip oil were expected to show peaks for both PVA and catnip, but surprisingly only peaks for PVA (2911 and 1418 cm$^{-1}$) were found. The subtraction spectrum, however, showed sharp peaks at 1660 and 814 cm$^{-1}$ as well as offset peaks at 2911 and 1437 cm$^{-1}$ which clearly matched the peaks of the pure catnip oil spectra, confirming the presence of catnip in the spun PVA nanofibrous structure.

## 3.4. Characterizing the repellency of the integrated mosquito repellents

Statistical analysis showed that for any tested concentration of micro-capsules, the number of mosquito landings reduced significantly compared to the control ($p \leq 0.001$). This showed an increase in repellency

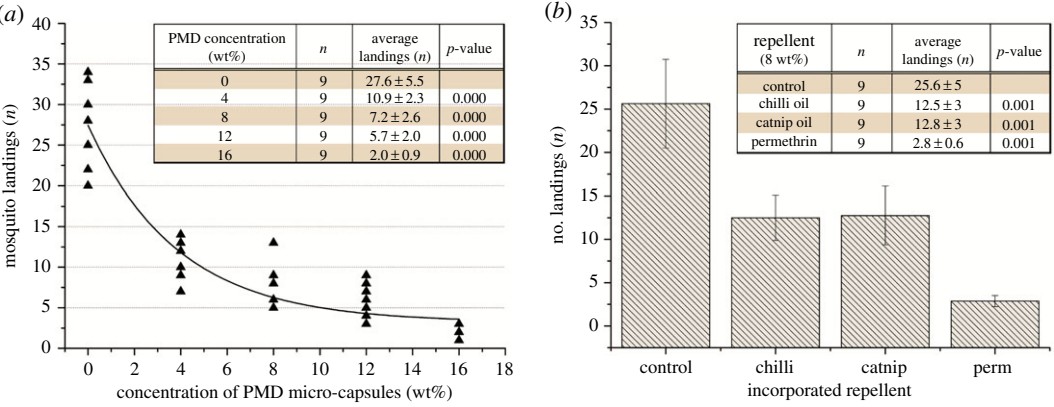

**Figure 6.** Number of landings by 10 female mosquitoes during 8 min observation time ($n = 9$). Two-sample Kolmogorov–Smirnov tests show the statistically significant repellency of: (*a*) different concentrations of PMD micro-capsules, (*b*) 8% chilli oil, catnip oil and permethrin incorporated in PVA nanofibrous structures [21].

against mosquitoes with increasing incorporation of PMD micro-capsules (figure 6*a*). The data showed an inverse exponential relationship between PMD concentration and the number of mosquito landings with a fitted equation of $y = 24.49\mathrm{e}^{-0.153x}$ and $R^2 = 0.85$.

However, the greatest decrease in mosquito landings occurred when the PMD concentration increased from 0 to 8% with higher concentrations only leading to a marginal decrease in the number of mosquito landings. To be able to compare the effectiveness of future different micro-encapsulated repellents in nanofibrous PVA mats, we calculated the halve value, which is the concentration of micro-capsules of a certain repellent needed to halve the amount of mosquito landings. In our case, a concentration of 3.75% micro-capsules of PMD was needed to halve the number of mosquito landings (value $Y = (27.6/2) = 13.8$).

To test the repellency of the nanofibrous structure spun with incorporated permethrin, chilli and catnip oil, a sample incorporated with 8% of each repellent were tested against malaria mosquitoes (*A. gambiae s.s.*). The results presented in figure 6*b* showed that all the incorporated repellents significantly reduced the number of mosquito landings compared to the control. The sample incorporated with permethrin had a significantly stronger effect than samples with chilli and catnip oil ($p < 0.01$). Incorporating chilli and catnip oil in PVA nanofibres reduced the amount of mosquito landings with 51%, while permethrin reduced the landing with 89% compared to the control.

## 4. Discussion

Recently, electrospinning has become popular for its versatility in fabrication of nanofibres from various synthetic and natural polymers. The special properties of electrospun nanofibrous structure, such as a relatively large surface-area-to-volume ratio and small pore sizes, make them favourable in various application areas like drug delivery, filtration, wound dressing, nanocomposites, etc. [7]. With the current focus on electrospinning, many innovations are made to the electrospinning set-up in an effort to produce functional nanofibrous structure with unique properties [22–25]. A new innovation is the incorporation of functional additives in the fibre matrix or fibre surface through electrospinning of micro-capsules in the nanofibres mats [26]. This technique provides long-term functionality; however, micro-capsules still need to be actively broken to release the functional additives. Another innovation is emulsion electrospinning which has become a method of choice due to its simplicity in the production of core–shell bi-component nanofibrous structures [14,27]. In emulsion electrospinning, functional additives are encapsulated and incorporated into nanofibres in a one-step process, making this technique simple and cheap [14]. Moreover, in contrast with micro-capsules, the encapsulated additives do not need active breaking for releasing the additives as they are continuously released through the thin wall of the fibre, which might result in shorter term use [14,26]. Therefore, both techniques have their pros and cons and here, we compared both techniques in the light of repellent integration and efficacy.

Incorporating additives into the electrospinning solutions can significantly affect the morphological and mechanical properties of the electrospun nanofibres and structures [24,26]. In this study, adding

PMD micro-capsules did not affect the fibre morphology (figure 2b). The fibre morphology of the electrospun samples of the reference blank PVA nanofibrous structure (figure 2a) looked similar to the sample with added PMD micro-capsules (figure 2b). The micro-capsules were covered by the nanofibres integrating them into the fabric, not changing the nanofibre morphology. This indicates that the spinning process was not affected by these micro-capsules. By contrast, nanofibres spun from emulsions with permethrin and catnip oil showed bead-like structures along the fibre axis (figure 2d,e), whereas fibres loaded with chilli oil resulted in porous fibres (figure 2f). The observed pores may have been a result of a bad emulsion, while the beads may suggest that repellents were encapsulated in the nanofibres during spinning as previously described in [1,27]. During emulsion electrospinning, jets are generated from the emulsion liquid, the dispersed drop in the emulsion makes the core of the electrospun fibres, while the continuous polymer matrix (here PVA) forms the shell [14,27]. This phenomenon is explained by the fact that water evaporates faster than oil whereby the polymer (PVA) solidifies faster than the oil drop (repellent) forming a shell around the oil drop. Additionally, the viscosity difference between the oil drop and the matrix may have led to an inward movement of the drop resulting into this core–sheath fibre [27,28].

Upon addition of the repellents PMD and permethrin, a significant effect was noted in the fibre diameters of the nanofibres ($p$-value: 0.0001), resulting in a decreasing fibre diameter with increasing concentration of PMD and permethrin (figure 3a,b). Contrarily, the diameter of the nanofibres spun with chilli and catnip oil significantly increased ($p < 0.01$) with increasing concentration of the repellents (figure 3c,d). This may be explained by the fact that incorporating these repellents during spinning introduced beads, voids and pores inside the fibres, reducing the density and increasing the volume of the nanofibres which resulted in the observed increase in the fibre diameter [27,29–31].

The selection of a textile material for various applications depends on its mechanical properties which determine the ability of the material to resist external damage and therefore also their end-use [32,33]. Therefore, the mechanical properties of the micro-capsule-nanofibrous structures were investigated to determine the quality of the resulting nanofibrous structure when compared with the blank PVA nanofibrous structure in terms of tensile strength, Young's modulus and elongation at break (figure 4). From our results, it was observed that introducing PMD micro-capsules, permethrin, chilli and catnip oil did not have a significant effect on all tested tensile properties of the resulting nanofibrous structure (figure 4). Therefore, the added repellents did not offer any reinforcement towards the PVA nanofibres matrix, neither did they introduce early stress failures which could have deteriorated the properties. This further implies that there was almost no interaction between the PVA nanofibres and the added repellents. For incorporated PMD micro-capsules, this suggests that these were located in the spaces between the fibres in the nanofibrous structure which can be seen in figure 2b. However, the standard deviations increased with increasing micro-capsule concentration which may indicate that more and more micro-capsules would fill up the available spaces in the nanoweb, suggesting that higher concentrations of micro-capsules would ultimately lead to significant effects in the mechanical properties [14,26].

Because Raman can give information about the polymer structure, the structure of guest molecules incorporated in the polymer matrix and even evaluate qualitatively the interaction between guest molecules and the polymer matrix [34–36], this vibrational spectroscopy method was used for determining the presence of the incorporated repellents. The spectra of the sample of PVA spun with each repellent showed peaks for both PVA and the respective repellents, demonstrating that all repellents were incorporated into the PVA nanofibrous structures (figure 5).

Furthermore, all resulting nanofibrous structures incorporated with the repellent greatly reduced the number of mosquito landings compared to the control (ranging from 50 to 93%). For the same concentrations, permethrin performs better (89% reduction) in repelling mosquitos than the other repellents like chilli (51% reduction), PMD (74% reduction) and catnip oil (50% reduction) [2,37]. Moreover, a potent repellent effect was observed in permethrin as is reported in previous studies [2,38,39].

These results clearly indicate that textile materials with incorporated novel repellents can be used in the protection against mosquitoes. These biological repellents may offer an alternative to the currently treated textiles with toxic insecticides [8,40]. Recently, biological repellents are considered as a tool in vector control preventing mosquito house entry [30,41], providing a promising application for our novel repelling textiles in push–pull system [42]. In the light of these potential applications, future studies should focus on the durability of the material, such as their washing durability and the longevity of the repellency efficacy. Also, as PVA is water soluble, it will be important to look at possible cross-linking methods to make the PVA added with the different repellents water-insoluble and independent of environmental humidity.

To conclude, mosquito repellents were successfully spun into PVA nanofibres with electrospinning to obtain a slow-release mosquito repellent textile. Two techniques were used, namely electrospinning of micro-capsules with incorporated repellent and direct emulsion electrospinning of the repellents. These methods both proved very successful in developing new and very effective mosquito-repelling textile material that can be used in the prevention against mosquito-associated diseases. Whereas, incorporation of micro-capsules in the nanofibres gives long-term functionality, it still needs active breakage of the micro-capsules to release the functional additive. Here, we only tested PMD, but it is obvious that these micro-capsules can contain many different repellents and even mixtures, making this an attractive, quick method for developing a variety of repelling textiles. On the other hand, emulsion electrospinning encapsulates the functional additives in a one-step process into nanofibres, making it a simple and cheap technique. Moreover, because the encapsulating wall is so thin, it may be possible that the additives are continuously released slowly, and it is therefore needless to break the nano-capsules to release the additive. However, this continuous release may result in short-term use. Both methods show great potential in creating novel and very efficient repelling textiles, not only on a laboratory-scale but also for industrial-scale productions.

Data accessibility. Data for this research work are available within the Dryad Digital Repository: https://doi.org/10.5061/dryad.jj72851 [21].

Authors' contributions. L.C. worked on electrospinning and compiled the paper. L.B. and J.M. sourced and prepared the repellents. L.V.L. tested the mechanical properties of the electrospun materials. D.M. and M.H. tested the repellency of the electrospun materials. L.V.L. was the overall research supervisor and adviser. K.D.C. supervised and advised on the electrospinning experiments. T.G. did the scanning electron microscopy and the Raman spectroscopy.

Competing interests. We declare we have no competing interests.

Funding. This work was done in the framework of the NOBUG project funded by European Commission, Grant agreement no. NMP2-SE-2009-228639.

Acknowledgements. We thank Devan Chemicals and Utexbel for supplying us with the mosquito repellents. We would also like to thank Katrien, Bert De Schoenmaker and Sarah Loomans for their technical help.

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
