## [Reviewer comments · Royal Society Open Science]

Review History

RSOS-182139.R0 (Original submission)

Review form: Reviewer 1

Is the manuscript scientifically sound in its present form?

No

Are the interpretations and conclusions justified by the results?

No

Is the language acceptable?

No

Is it clear how to access all supporting data?

Not Applicable

Do you have any ethical concerns with this paper?

No

Have you any concerns about statistical analyses in this paper?

I do not feel qualified to assess the statistics

Recommendation?

Reject

Comments to the Author(s)

Ciera and coworkers present an interesting paper on the efficacy of repellent-treated fabrics. Unfortunately, the authors don't seem familiar with the current literature and make many unsubstantiated claims in the introduction and discussion. Some mayor revisions would be required before this paper could be published.

Introduction:

Page 2, Line 13: permethrin is a pyrethroid class insecticide with repellent properties. It is different from the other repellents that can be sprayed on peoples skins.

Line 14-16 The citations used to show that 'synthetic repellents' are toxic to humans are not valid. There is little in the literature to suggest that for example DEET is more toxic to humans like for example PMD. The authors should take a critical look at the literature and rewrite their introduction.

Line 19-20 Better say that there is a large market for 'alternative repellents'.

Line 6-42 Many of the citations in the introduction (like for example Science Daily) are not acceptable and need to be replaced with scientific paper citations.

Materials - should be Materials & Methods

Page 5 Line 27 - please describe the species of mosquitoes used.

Line 51 the bioassay is not well described. Please add a scheme and photo so the reader can understand how this assay works.

Page 6 Line 6 - Provide details of the solution from Mukabana and add citation.

Discussion:

Many of the citations are simply missing

Page 16 line 48 the authors contrast synthetic repellents with biological repellents. This is unacceptable language.

Line 52 none of the repellents tested here are novel.

Line 53 Permethrin-treated clothing is already on the market.

Page 17 Line 10 Is there any proof that the textiles actually slowly release the repellents? If not these claims must be removed.

Review form: Reviewer 2**Is the manuscript scientifically sound in its present form?**

Yes

Are the interpretations and conclusions justified by the results?

Yes

Is the language acceptable?

Yes

Is it clear how to access all supporting data?

Not Applicable

Do you have any ethical concerns with this paper?

No

Have you any concerns about statistical analyses in this paper?

No

Recommendation?

Accept with minor revision (please list in comments)

Comments to the Author(s)

Comments to the editor

This paper reports micro-encapsulation of biological mosquito repellents into nanofibers through electrospinning for controlled slow release. This reviewer finds the paper to have interesting technical contents and is likely to attract a wider readership and therefore recommends its publications after some minor revisions.

Comments to the author

Why were the mosquito nanofabrics not tested in the field because this could have given a clear perspective of the application of these innovative textiles?

Generally the author has mixed up present and past tense- this need to be revised

There seems to be some grammatical errors that need to be corrected

There is a grammatical error in the abstract- last sentence of paragraph one

For example page 2 line26-28

Page 7 line 44 - replace "is" and "smaller" with "was" and "less", respectively.

Page 10 line 18 - replace "is" with "was"

Page 11 line 49 "shows " should be "showed" same case for page 11 line 53

Page 12 line 18,22,27- revise

Page 16 line 26

Page 16 line 45- delete "by"

Page 17 line 6-10- revise

Decision letter (RSOS-182139.R0)

24-Apr-2019

Dear Dr Ciera,

The editors assigned to your paper ("Electrospinning repellents in polyvinyl alcohol-nanofibers for obtaining mosquito repelling fabrics") have now received comments from reviewers. We would like you to revise your paper in accordance with the referee and Associate Editor suggestions which can be found below (not including confidential reports to the Editor). Please note this decision does not guarantee eventual acceptance.

Please submit a copy of your revised paper before 17-May-2019. Please note that the revision deadline will expire at 00.00am on this date. If we do not hear from you within this time then it will be assumed that the paper has been withdrawn. In exceptional circumstances, extensions may be possible if agreed with the Editorial Office in advance. We do not allow multiple rounds of revision so we urge you to make every effort to fully address all of the comments at this stage. If deemed necessary by the Editors, your manuscript will be sent back to one or more of the

original reviewers for assessment. If the original reviewers are not available, we may invite new reviewers.

- Data accessibility

If you wish to submit your supporting data or code to Dryad (<http://datadryad.org/>), or modify your current submission to dryad, please use the following link:
<http://datadryad.org/submit?journalID=RSOS&manu=RSOS-182139>

- Competing interests

- Authors' contributions

- Acknowledgements

- Funding statement

Kind regards,

Andrew Dunn

on behalf of Prof R. Kerry Rowe (Subject Editor)

Associate Editor's comments:

Two reviewers have commented on your manuscript, and each recommend revisions - in the case of the first reviewer, these are substantial. You should make every effort to tackle these concerns, and provide a full point-by-point response when you resubmit. If you are not able to persuade the reviewers that the revised manuscript should be accepted, we will not be able to consider the manuscript further.

Comments to Author:

Reviewers' Comments to Author:

Reviewer: 1

Comments to the Author(s)

Ciera and coworkers present an interesting paper on the efficacy of repellent-treated fabrics. Unfortunately, the authors don't seem familiar with the current literature and make many unsubstantiated claims in the introduction and discussion. Some major revisions would be required before this paper could be published.

Introduction:

Page 2, Line 13: permethrin is a pyrethroid class insecticide with repellent properties. It is different from the other repellents that can be sprayed on people's skins.

Line 14-16 The citations used to show that 'synthetic repellents' are toxic to humans are not valid. There is little in the literature to suggest that for example DEET is more toxic to humans like for example PMD. The authors should take a critical look at the literature and rewrite their introduction.

Line 19-20 Better say that there is a large market for 'alternative repellents'.

Line 6-42 Many of the citations in the introduction (like for example Science Daily) are not acceptable and need to be replaced with scientific paper citations.

Materials - should be Materials & Methods

Page 5 Line 27 - please describe the species of mosquitoes used.

Line 51 the bioassay is not well described. Please add a scheme and photo so the reader can understand how this assay works.

Page 6 Line 6 - Provide details of the solution from Mukabana and add citation.

Discussion:

Many of the citations are simply missing

Page 16 line 48 the authors contrast synthetic repellents with biological repellents. This is unacceptable language.

Line 52 none of the repellents tested here are novel.

Line 53 Permethrin-treated clothing is already on the market.

Page 17 Line 10 Is there any proof that the textiles actually slowly release the repellents? If not these claims must be removed.

Reviewer: 2

Comments to the Author(s)

Comments to the editor

This paper reports micro-encapsulation of biological mosquito repellents into nanofibers through electrospinning for controlled slow release. This reviewer finds the paper to have interesting technical contents and is likely to attract a wider readership and therefore recommends its publications after some minor revisions.

Comments to the author

Why were the mosquito nanofabrics not tested in the field because this could have given a clear perspective of the application of these innovative textiles?

Generally the author has mixed up present and past tense- this need to be revised

There seems to be some grammatical errors that need to be corrected

There is a grammatical error in the abstract- last sentence of paragraph one

For example page 2 line26-28

Page 7 line 44 - replace "is" and "smaller" with "was" and "less", respectively.

Page 10 line 18 - replace "is" with "was"

Page 11 line 49 "shows " should be "showed" same case for page 11 line 53

Page 12 line 18,22,27- revise

Page 16 line 26

Page 16 line 45- delete "by"

Page 17 line 6-10- revise

Author's Response to Decision Letter for (RSOS-182139.R0)

See Appendix A.

RSOS-182139.R1 (Revision)

Review form: Reviewer 2

Is the manuscript scientifically sound in its present form?

Yes

Are the interpretations and conclusions justified by the results?

Yes

Is the language acceptable?

Yes

Is it clear how to access all supporting data?

Yes

Do you have any ethical concerns with this paper?

No

Have you any concerns about statistical analyses in this paper?

No

Recommendation?

Accept as is

Comments to the Author(s)

The authors of this work have made satisfactory corrections and the current reviewer recommends the publication of this manuscript as is.

Review form: Reviewer 3

Is the manuscript scientifically sound in its present form?

Yes

Are the interpretations and conclusions justified by the results?

Yes

Is the language acceptable?

Yes

Is it clear how to access all supporting data?

Yes

Do you have any ethical concerns with this paper?

No

Have you any concerns about statistical analyses in this paper?

I do not feel qualified to assess the statistics

Recommendation?

Accept with minor revision (please list in comments)

Comments to the Author(s)

The manuscript reports a preparation of mosquito repelling nanofibrous fabric by electrospinning repellents-containing polymer solution. It is interesting and of practically useful of the functionalized material by a facile technique. During the revision, most of the comments are well replied. However, this reviewer would suggest a minor revision with several additional questions:

- 1) Since the used mosquito repellents are volatile that will be easily evaporated during electrospinning, so is it possible to verify the exact amount of repellents remaining in the nanofiber after electrospinning?
- 2) PAV is a water-soluble polymer, as claimed by the authors it is environmental; however, just because it is water-soluble it would be easily dissolved for the resulting PVA nanofibers since applying environment is moist/wet/high humidity, and for human skin, it also might be sweating. So, how to address this issue?
- 3) SEM image of nanofiber containing different amount of repellents should be provided.
- 4) The size of the microcapsules is $5 \pm 1 \mu\text{m}$, how does such a big size particles being encapsulated in nanofibers with the diameter of about several hundreds nanometers?
- 5) In the experiment section, it is written that the PVA solution was prepared under stirring at room temperature ($21 \pm 1^\circ\text{C}$), but as I know the PVA could usually be dissolved under heating ($\sim 90^\circ\text{C}$). Can you answer this question?
- 6) What is the thickness of the nanofiber mat used for the mechanical test (both the directly measured and calculated ones)?

Review form: Reviewer 4

Is the manuscript scientifically sound in its present form?

Yes

Are the interpretations and conclusions justified by the results?

Yes

Is the language acceptable?

Yes

Do you have any ethical concerns with this paper?

No

Recommendation?

Accept as is

Comments to the Author(s)

The authors have addressed all the comments appropriately from the reviewers and the work of using electrospun fibers loaded with repellents to repel mosquito is very interesting and can be useful.

Decision letter (RSOS-182139.R1)

04-Jul-2019

Dear Dr Ciera:

On behalf of the Editors, I am pleased to inform you that your Manuscript RSOS-182139.R1 entitled "Electrospinning repellents in polyvinyl alcohol-nanofibers for obtaining mosquito repelling fabrics" has been accepted for publication in Royal Society Open Science subject to minor revision in accordance with the referee suggestions. Please find the referees' comments at the end of this email.

The reviewers and Subject Editor have recommended publication, but also suggest some minor revisions to your manuscript. Therefore, I invite you to respond to the comments and revise your manuscript.

- Ethics statement

- Data accessibility

If you wish to submit your supporting data or code to Dryad (<http://datadryad.org/>), or modify your current submission to dryad, please use the following link:
<http://datadryad.org/submit?journalID=RSOS&manu=RSOS-182139.R1>

- Competing interests

- Authors' contributions

- Acknowledgements

- Funding statement

Because the schedule for publication is very tight, it is a condition of publication that you submit the revised version of your manuscript before 13-Jul-2019. Please note that the revision deadline will expire at 00.00am on this date. If you do not think you will be able to meet this date please let me know immediately.

on behalf of R. Kerry Rowe (Subject Editor)
openscience@royalsociety.org

Associate Editor Comments to Author:

Thank you for submitting your revision and please accept our apologies for the length of time the revision has been under review: unfortunately, one of the original reviewers was not available to assess your revision, necessitating the journal find additional/new reviewers. This process took somewhat longer than we would have preferred.

Nevertheless, the reviewers are now of the view that your paper is improved, though one of the new reviewers has identified a number of minor points that you must address in your revision and point-by-point response. We'll look forward to receiving these shortly.

Reviewer comments to Author:
Reviewer: 2

Comments to the Author(s)
The authors of this work have made satisfactory corrections and the current reviewer recommends the publication of this manuscript as is.

Reviewer: 3

Comments to the Author(s)
The manuscript reports a preparation of mosquito repelling nanofibrous fabric by electrospinning repellents-containing polymer solution. It is interesting and of practically useful of the functionalized material by a facile technique. During the revision, most of the comments are well replied. However, this reviewer would suggest a minor revision with several additional questions:

- 1) Since the used mosquito repellents are volatile that will be easily evaporated during electrospinning, so is it possible to verify the exact amount of repellents remaining in the nanofiber after electrospinning?
- 2) PAV is a water-soluble polymer, as claimed by the authors it is environmental; however, just because it is water-soluble it would be easily dissolved for the resulting PVA nanofibers since

applying environment is moist/wet/high humidity, and for human skin, it also might be sweating. So, how to address this issue?

3) SEM image of nanofiber containing different amount of repellents should be provided.

4) The size of the microcapsules is $5 \pm 1 \mu\text{m}$, how does such a big size particles being encapsulated in nanofibers with the diameter of about several hundreds nanometers?

5) In the experiment section, it is written that the PVA solution was prepared under stirring at room temperature ($21 \pm 1^\circ\text{C}$), but as I know the PVA could usually be dissolved under heating ($\sim 90^\circ\text{C}$). Can you answer this question?

6) What is the thickness of the nanofiber mat used for the mechanical test (both the directly measured and calculated ones)?

Reviewer: 4

Comments to the Author(s)

The authors have addressed all the comments appropriately from the reviewers and the work of using electrospun fibers loaded with repellents to repel mosquito is very interesting and can be useful.

Author's Response to Decision Letter for (RSOS-182139.R1)

See Appendix B.

Decision letter (RSOS-182139.R2)

24-Jul-2019

Dear Dr Ciera,

I am pleased to inform you that your manuscript entitled "Electrospinning repellents in polyvinyl alcohol-nanofibers for obtaining mosquito repelling fabrics" is now accepted for publication in Royal Society Open Science.

Best regards,

on behalf of R. Kerry Rowe (Subject Editor)
openscience@royalsociety.org

Follow Royal Society Publishing on Twitter: [@RSocPublishing](https://twitter.com/RSocPublishing)

Appendix A

9th May 2019

To The Editor

Dear Andrew Dunn,

RE: Manuscript ID RSOS-182139 revision:

Thank you for allowing us to resubmit a revised version of our manuscript entitled, "*Electrospinning repellents in polyvinyl alcohol-nanofibers for obtaining mosquito repelling fabrics*" for publication consideration in your Journal (**Royal Society Open Science**).

We appreciate the reviewer comments, which we have carefully considered. An outline of the reviewers' comments and our changes and/or rebuttal is described below. Our responses to the reviewer's comments appear in italic. Thank you for your continued assistance throughout this process.

1st Review

Introduction:

Page 2, Line 13: permethrin is a pyrethroid class insecticide with repellent properties. It is different from the other repellents that can be sprayed on peoples skins.

We didn't indicate that permethrin is the same with other repellents that can be applied on the skin. We just mentioned it as an example of a synthetic repellent. Please see the paragraph below:

The most effective and most commonly used insect repellents for preventing mosquito bites present on the market today are synthetic in nature which include DEET (N, N-diethyl-3-methylbenzamide), permethrin and Picaridin [2, 4, 5]. However, some of these synthetic repellents are associated with some harmful and toxic effects, some resulting in allergic reactions and damage to the nervous system [6, 7]. Moreover, some mosquitoes have become resistant to the current synthetic repellents and environmental concerns have prevented their effective use [3, 5]

Line 14-16 The citations used to show that 'synthetic repellents' are toxic to humans are not valid. There is little in the literature to suggest that for example DEET is more toxic to humans like for example PMD. The authors should take a critical look at the literature and rewrite their introduction.

We were not comparing the toxicity of the different repellents. We just mentioned that some synthetic repellents can be harmful. Please see below

However, some of these synthetic repellents are associated with some harmful and toxic effects, some resulting in allergic reactions and damage to the nervous system [6, 7]. Moreover, some

mosquitoes have become resistant to the current synthetic repellents and environmental concerns have prevented their effective use [3, 5].

Line 19-20 Better say that there is a large market for 'alternative repellents'.

Corrected. Please See below

Therefore, there is a large market for alternative repellents. Thus, lemon eucalyptus, chili and catnip oil were used in this study beside Permethrin. P-menthane-3, 8-diol (PMD) is the active compound of lemon eucalyptus oil and is extracted from the twigs and leaves of *Eucalyptus citriodora* of the Myrtaceae family [5, 7].

Line 6-42 Many of the citations in the introduction (like for example Science Daily) are not acceptable and need to be replaced with scientific paper citations.

Corrected. More scientific articles were added

Materials - should be Materials & Methods

Corrected

Page 5 Line 27 - please describe the species of mosquitoes used.

Corrected and added more text. Please See below

Mosquitoes: *Anopheles gambiae s.s.* were used in this study. *A. gambiae s.s.* a member of the homonymous species complex comprising six recognized sibling species, is the most anthropophilic malaria vector worldwide [16, 17]. Within the *Anopheles gambiae* (Diptera: Culcidae) complex *An. gambiae sensu stricto (s.s)* is the most specifically adapted to humans and has the highest malaria vectorial capacity [18, 19].

Line 51 the bioassay is not well described. Please add a scheme and photo so the reader can understand how this assay works.

Corrected. Please See below

Figure 1. Schematic representation of the repellent bioassay, showing the flight chamber containing the assay cage and the position of the circular landing platform (arrow) emitting five-component attractant blend and moisture, on which the repellent-impregnated PVA nanofibers [20]

Page 6 Line 6 - Provide details of the solution from Mukabana and add citation.

The description is long and will make the manuscript too long instead we have added the citation. Please below

The strips were then impregnated with attractive compounds by bringing them into an Eppendorf tube containing 1 mL of solution as described in [21]. Thereafter, the strips were stored overnight in a refrigerator at 4°C and later hung up for half an hour under a fume hood to allow excess fluids to leak out. Finally, the strips were immediately used or packed in aluminium foil and stored at 4°C in a refrigerator until they were used.

Many of the citations are simply missing

Corrected. The missing citations were added

Page 16 line 48 the authors contrast synthetic repellents with biological repellents. This is unacceptable language.

Corrected. The sentence was deleted. Please see below

For the same concentrations, Permethrin performs better (89% reduction) in repelling mosquitos than the other repellents like chili (51% reduction), PMD (74% reduction) and catnip oil (50% reduction), [2, 39]. Moreover, a potent repellent effect was observed in permethrin as is reported in previous studies [2, 40, 41].

Line 52 none of the repellents tested here are novel.

Our resulting Nano fabric is what we are referring to as novel not the repellents. Please see below

These results clearly indicate that the novel textile materials with incorporated biological repellents can be used in the protection against mosquitoes. Recently, biological repellents are considered as a tool in vector control preventing mosquito house entry [32, 42, 43], providing a promising application for our novel repelling textiles in push-pull system [44]. In light of these potential applications, future studies should focus on the durability of the material, such as their washing durability and the longevity of the repellency efficacy.

Line 53 Permethrin-treated clothing is already on the market.

Yes but none of the permethrin treated clothing already in the market is produced using the reported techniques, namely micro encapsulation nor emulsion electrospinning

Page 17 Line 10 Is there any proof that the textiles actually slowly release the repellents? If not these claims must be removed.

Corrected by indicating that there is possible slow release. Please see below

To conclude, mosquito repellents were successfully spun into PVA nanofibers with electrospinning to obtain a possible slow release mosquito repellent textile.

Reviewer: 2

Why were the mosquito nanofabrics not tested in the field because this could have given a clear perspective of the application of these innovative textiles?

Saly enough it was not possible to perform these very useful tests in the framework of this research but this will be done in the next phase of the study

There is a grammatical error in the abstract- last sentence of paragraph one

Corrected. Please see below

The focus of this study is on the direct micro-encapsulation of repellents from an emulsion and the integration of already encapsulated repellents into nanofibers via electrospinning.

For example page 2 line26-28

Corrected. Please see below

Lastly, microencapsulation prevents repellents from being in direct contact with the skin, therefore reducing skin irritation and/or allergic reactions. An additional advantage of electrospinning producing fibers at the nano scale level (Φ : 10 to 800nm) [13].

Page 7 line 44 – replace “is” and “smaller” with “was” and “less”, respectively.

Corrected. Please see below

A decreasing trend was seen in the diameter of the nanofibres upon adding more microcapsules, with 16% PMD microcapsules showing the less average diameter and a large size distribution (386 ± 55 nm) against the blank (408 ± 67 nm).

Page 10 line 18 – replace “is” with “was”

Corrected. Please see below

Additionally, there was a big variability in the fiber diameters for fibers loaded with chili in contrast to the others with increasing concentrations.

Page 11 line 49 “shows ” should be “showed” same case for page 11 line 53

Corrected. Please see below

Although not significant, samples incorporated with catnip did showed a slight decrease in mechanical properties.

The spectrum of blank PVA showed characteristic peaks at 848, 2906, 2911, 1437, 1438 and 1439 cm^{-1} (Figure 5 (black)).

Page 12 line 18,22,27- revise

Corrected. Please see below

For the pure permethrin spectra (Figure 5, b), four characteristic peaks at 3064, 1614, 1437 and 1000 cm^{-1} could be observed. When subtracting the spectra of pure PVA from PVA with permethrin, the same characteristic peaks, namely 3064, 1614 and 1000 cm^{-1} , were present like in pure permethrin (not found in PVA) and therefore can be used for the identification of permethrin in PVA nanofibrous structures. The spectra for the sample of PVA incorporated with permethrin showed the peaks for both PVA and permethrin, namely at 1614 and 1000 cm^{-1} for permethrin, and 1437 for PVA, confirming the presence of permethrine in the spun PVA nanofibrous structure.

Page 16 line 26

Corrected. Please see below

Statistical analysis showed that for any tested concentration of microcapsules, the number of mosquito landings reduced significantly compared to the control ($p\leq 0.001$).

Page 16 line 45- delete “by”

Corrected. Please see below

Furthermore, all resulting nanofibrous structures incorporated with the repellent greatly reduced the number of mosquito landings compared to the control (ranging from 50 to 93%).

Page 17 line 6-10- revise

Corrected. Please see below

To conclude, mosquito repellents were successfully spun into PVA nanofibers with electrospinning to obtain a possible slow release mosquito repellent textile. Two techniques were used, namely electrospinning of microcapsules with incorporated repellent and direct emulsion electrospinning of the repellents. These methods both proved very successful in developing new and very effective mosquito repelling textile material that can be used in the prevention against mosquito associated diseases. Whereas, incorporation of microcapsules in the nanofibers gives long term functionality it still needs active breakage of the microcapsules to release the functional additive. Here we only tested PMD, but it is obvious that these microcapsules can contain many different repellents and even mixtures, making this an attractive, quick method for developing a variety of repelling textiles. On the other hand, emulsion electrospinning encapsulates the functional additives in a one-step process into nanofibers making it a simple and cheap technique. Moreover, because the encapsulating wall is so thin, it may be possible that the additives are continuously released slowly and it is therefore needless to break the nano-capsules to release the additive. However, this possible continuous release may result in short term use. Both methods show great potential in creating novel and very efficient repelling textiles, not only on a lab-scale but also for industrial-scale productions.

Appendix B

10th July 2019

To The Editor

Dear Editor,

RE: Manuscript ID RSOS-182139.R1

We would like to thank you for your email and your continuous effort to improve our paper. We have read and answered the comments of the reviewers in great detail.

As reviewer 2 and 4 did not have any more questions/remarks we only could answer the remarks of reviewer 3 who had some valid points:

An outline of the reviewers' comments and our changes and/or rebuttal is described below. Our responses to the reviewer's comments appear in italic. Thank you for your continued assistance throughout this process.

- 1) Since the used mosquito repellents are volatile that will be easily evaporated during electrospinning, so is it possible to verify the exact amount of repellents remaining in the nanofiber after electrospinning?

This is a valid point and some of the repellent will evaporate during the electrospinning process, however we think this loss will be very minimal during the process for the following reasons:

- *The repellent is in emulsion in PVA and is oil like, therefore water from the PVA solution will still be evaporated first and in greater amounts than the repellent, although it is possible that some small quantities of the repellent could evaporate.*
- *We still see beads in the fibres compared to the blank meaning that the repellent did get included in the PVA nanofibrils (Figure 2)*
- *We showed that the repellents are present in the fibrous mats with infrared which is very clear (Figure 5)*
- *We can show a repellent effect on mosquitos of our fibrous mats which is a macroscopic effect of a nanofibrous mat, meaning that the repellent that is integrated into the electrospun mats must be significantly high (Figure 6)*

So in general, we are convinced that the repellent is integrated in the electrospun mats and in large quantities. In the future, we of course will quantify these amounts but it is extremely hard to also show this. As for now there is no straight forward technique to do this. So yes, this will be done in the future but is not part of the current paper.

- 2) PVA is a water-soluble polymer, as claimed by the authors it is environmental; however, just because it is water-soluble it would be easily dissolved for the resulting PVA nanofibers since applying environment is moist/wet/high humidity, and for human skin, it also might be sweating. So, how to address this issue?

Again, a very valid point and it is strange that this question has not been raised before. PVA is indeed water soluble and indeed in high humidity situations it is possible that the structure would dissolve. However, there are already many techniques known in the literature that show you can crosslink PVA to make it water insoluble (Affandi et al., 2012). These techniques are quite standard and are known to work so we see no problem on applying these to our fabrics to make them water insoluble.

*Therefore a sentence was added in the manuscript in the discussion section:
"Also, as PVA is water-soluble, it will be important to look at possible crosslinking methods to make the PVA in combination with the different repellents water-insoluble and independent of environmental humidity."*

- 3) SEM image of nanofiber containing different amount of repellents should be provided.
For this paper and its focus we do not see the added value of adding additional SEM image of nanofiber containing different amount of repellents. We think the given pictures (Figure 2) are sufficient and clear, and additional pictures in the manuscript would overload the paper with images and make it extra-long.
- 4) The size of the microcapsules is $5 \pm 1 \mu\text{m}$, how does such a big size particles being encapsulated in nanofibers with the diameter of about several hundred nanometers?
This is a strange question for someone in the field of electrospinning as this is quite basic. In short, in the paper we mentioned that we are using a needle of 15 cm long stainless steel needle with an inner diameter of 1.024 mm. This diameter of the needle is not the ultimate diameter of the spun fibres as these are formed with the Taylor cone and during evaporation. So we go from 1mm to hundreds of nanometers during the spinning process. It is thus clear that the microcapsules can clearly pass the needle and are expelled during the electrospinning process, forming an equal distribution in the mat during the spinning of the PVA.
- 5) In the experiment section, it is written that the PVA solution was prepared under stirring at room temperature ($21 \pm 1^\circ\text{C}$), but as I know the PVA could usually be dissolved under heating ($\sim 90^\circ\text{C}$). Can you answer this question?

It is true, both techniques work. Elevated temperatures will indeed dissolve the PVA faster and then you can use it sooner too. We opted for dissolving it at room temperature but for a much longer time in order to avoid heat damage to the PVA.

- 6) What is the thickness of the nanofiber mat used for the mechanical test (both the directly measured and calculated ones)?
The thickness of the mats was not taken as it is extremely difficult to measure this accurately. Instead we considered the width and the length of the sample (3 x 30 mm) sample strips. Afterwards all samples were weighed having an average weight of $7,71 \pm 1 \text{ mg}$ to ensure an equal thickness between samples. This leads to a much better comparison between samples. (This is explained in the material and method section)."